:ᐰ: PLOS | ONE

# Nutritional inadequacies in commercial vegan foods for dogs and cats

**Rafael Vessecchi Amorim Zafalon[1], Larissa Wünsche Risolia[1], Thiago Henrique Annibale Vendramini[1], Roberta Bueno Ayres Rodrigues[1], Vivian Pedrinelli[1], Fabio Alves Teixeira[1], Mariana Fragoso Rentas[1], Mariana Pamplona Perini[1], Isabella Corsato Alvarenga[2], Marcio Antonio Brunetto(ID)[1]***

**1** School of Veterinary Medicine and Animal Science, University of São Paulo, São Paulo, Pirassununga, Brazil, **2** Department of Grain Science and Industry, Kansas State University, Manhattan, Kansas, United States of America

\* mabrunetto@usp.br

**Data Availability Statement:** All relevant data are within the manuscript and its Supporting Information files.

## Abstract

The objective of this study was to evaluate the macronutrients composition, fatty acid and amino acid profiles, and essential minerals content of all vegan foods for dogs and cats available in the Brazilian market, and to compare results with FEDIAF (2019) and AAFCO (2019) recommendations. Four vegan pet foods were assessed (three for dogs and one for cats). The comparisons were made in a descriptive manner. All foods met the minimum recommendations for macronutrients. Arachidonic acid was not reported in any food label. Regarding the FEDIAF recommendations, one food for dogs had low calcium, another had low potassium and a third had low sodium. The cat food presented potassium content lower than recommended. The Ca:P ratio did not meet the minimum recommendation of FEDIAF (2019) and AAFCO (2019) in any of the dog's foods analyzed, and the cat food also did not present the minimum recommendation based on FEDIAF (2019). Copper concentrations exceeded the legal limit in all foods. Zinc concentrations exceeded this limit in two foods (one for dogs and one for cats) and iron levels exceeded the legal limit in one dog food. One of the dog foods did not meet the minimum recommendation for methionine and the cat food did not meet the minimum recommendation for arginine. In addition, when the amount of nutrients consumed by animals with low energy requirements was simulated, in addition to the same non-conformities described above, it was observed that the cat food does not meet the minimum recommended of protein and taurine in unit/$Kg^{0.67}$. It was concluded that all foods analyzed had one or more nutrients below the recommended levels and some presented zinc and copper excess, therefore, these foods should not be recommended for dogs and cats, because dietary deficiencies found may lead to health risks for dogs and cats. Furthermore, manufacturers should review their formulations to ensure the nutritional adequacy of these foods.

**Funding:** The author(s) received no specific funding for this work.

**Competing interests:** The authors have declared that no competing interests exist.

## Introduction

Veganism is considered a strict form of vegetarianism and involves practices other than changes in the dietary regimen. Adepts of this movement exclude meat, seafood, eggs, dairy products, and all other animal products from their diet. Among the reasons that lead to the adoption of this lifestyle are health concerns, sustainability/environmental preservation, empathy for animals and ethical concerns [1].

According to a survey, around 14% of Brazilians declare themselves as vegetarian or vegan and this number seems to be growing [2]. The ethical appeal of veganism is also reflected in the search for foods that are free of animal origin ingredients for pets, which can have negative implications on animal health, especially in cats that are obligate carnivores. In spite of nutritional challenges, the number of supporters of pet vegan foods has increased [1], as well as the number of commercial diets of this segment commercially available.

Considering the physiology of dogs and cats, which classifies them respectively as facultative and obligate carnivores, the adequacy of vegetarian / vegan diets in supplying the minimum requirement of essential nutrients is questionable. The proper formulation of this type of food is a challenge since several of these nutrients are found mainly in ingredients of animal origin. A previous study found that only two out of 12 commercial vegetarian foods met the recommendations of the species for which they were intended [3].

The increase of adherents to veganism has stimulated the development of nutritionally complete and balanced plant-based diets that meet both the minimal nutritional requirements of dogs and cats, and vegan ethical constraints. Almost half of pet owners interested in providing plant-based diets seek more information to ensure the nutritional adequacy of these foods [1], which demonstrates their concern about the pet food quality. Nevertheless, only few studies have evaluated the composition of vegan foods or the impact of these diets on pet health. Thus, the objective of this study was to evaluate the macronutrient composition, fatty acids and amino acids profile, and essential minerals contents of vegan pet foods available in the Brazilian market, and to assess their compliance to recommended allowances for dogs and cats.

## Material and methods

### Sample selection

A survey was conducted to gather all vegan commercial dry extruded pet foods registered in Brazil. First, the key words "vegan" and "pet food" were used to search all foods that fit into this category in the Brazilian ministry of agriculture database. Next, a telephone survey with the major pet food distributors in the country was carried out to confirm these foods were available. Based on this survey, there were only four dry extruded vegan pet foods commercially available (three for dogs and one for cats) in the Brazilian market. A bag of each vegan pet food brand was purchased (ranging from 3 to 15 kg) at a local store. The nutritional composition of each food is described in Table 1.

### Nutrient analyses

**Proximate analyses.** The four foods were ground in a hammermill equipped with a 1 mm sieve. Dry matter was determined in an oven at 105°C (Fanen 315, São Paulo, Brazil) for eight hours according to AOAC [4]. Crude protein (CP) analysis was performed by the Kjeldahl method [4]. The ether extract was performed by the Soxhlet method after acid hydrolysis, and ash was determined using a muffle furnace at 550°C [4]. The crude fiber was determined

**Table 1. Ingredient composition declared on the label of the vegan pet foods analyzed in this study.**

| Foods | Ingredients |
|---|---|
| **Food A** | Organic whole wheat flour, organic soybean meal, organic soybean oil, yeast extract, brewer's yeast, sugar cane fiber, chickpea, organic brown rice flour, sugar cane fiber, wheat gluten, organic flaxseed, organic chia, dehydrated carrot, olive oil, dicalcium phosphate, bentonite, vitamins (A, D, E, K, C, B1, B2, B12, biotin, calcium pantothenate, folic acid, niacin, choline chloride), minerals (iron sulfate, copper sulfate, calcium iodate, potassium chloride, zinc oxide, manganese oxide, sodium selenite), methionine, plant extracts (yucca, rosemary, green tea and mint; 0.06%). |
| **Food B** | Broken rice, whole corn, corn gluten meal, soy bran, wheat bran, hydrolyzed yeast, yeast cell wall, yucca extract (0.025%), flax flour, probiotic additive (0.01%), prebiotic additive (0.02%), beet pulp, sodium chloride, degummed soybean oil (7.5%), hydrogenated soybean oil, folic acid, pantothenic acid, nicotinic acid, biotin, choline chloride, calcium iodate, pyridoxine, riboflavin, sodium selenite, cobalt sulphate, copper sulphate, dicalcium phosphate, calcium carbonate, manganese sulphate, magnesium sulphate, zinc sulphate, ferrous sulphate, vitamin A, vitamin B12, vitamin D2, vitamin E, vitamin K, l-lysine, dl-methionine, natural dye of annatto (0.06%), bentonite, dolomite, algae meal, potassium sorbate, propionic acid and natural antioxidant (0.05%) (tocopherol, silicon dioxide, rosemary oil, citric acid). |
| **Food C** | Ground whole maize, wheat bran, soybean meal, broken rice, corn gluten meal-60, brewer's dry yeast, yeast extract, degummed soybean oil (7%: omega 3 0 - .055% and omega 6–3.64%), calcitic limestone, dicalcium phosphate, sodium chloride (common salt), potassium chloride, vehicle q.s.p, natural antioxidant (0.06%—vegetable oil, citric acid, silicon dioxide, tocopherol extract, rosemary essence), yucca extract (0.0625%), propionic acid, calcium sodium aluminosilicate, potassium sorbate, flavour additive (dried sugar cane yeast, dried brewer's yeast, dextrose, phosphoric acid, potassium sorbate, tocopherol, rosemary oil), l-lysine, dl-methionine, zinc oxide, zinc amino acid chelate, copper sulfate, iron sulfate, manganese monoxide, calcium iodate, sodium selenite, organic selenium, choline chloride, folic acid, nicotinic acid, biotin, calcium pantothenate, vitamin A, vitamin B1, vitamin B12, vitamin B6, vitamin D3, vitamin E, vitamin K3, sodium hexametaphosphate, mannanoligosaccharide (0.2%), bacillus subtilis (0.038%). |
| **Food D** | Ground whole maize, corn gluten meal, soy hull, broken rice, pea protein, wheat bran, hydrolyzed yeast, yeast cell wall, yucca extract (0.025%), flax flour, (0.5%), probiotic additive (0.01%), prebiotic additive (0.02%), beet pulp, sodium chloride, dehydrated soybean oil (7.5%), hydrogenated soybean oil, folic acid, pantothenic acid, nicotinic acid, biotin, choline chloride, calcium iodate, pyridoxine, riboflavin, vitamin B1, sodium selenite, cobalt sulphate, copper sulphate, dicalcium phosphate, calcium carbonate, manganese sulphate, magnesium sulphate, zinc sulphate, ferrous sulphate, vitamin A, vitamin B12, vitamin D2, vitamin E, vitamin K, l-lysine, dl-methionine, l-carnitine, taurine (0.15%), natural dye of annatto %, bentonite, dolomite, algae meal, potassium sorbate, phosphoric acid, citric acid, propionic acid and natural antioxidant (0.05%) (tocopherol, silicon dioxide, rosemary oil, citric acid). |

according to the Weende method [5]. All analyses were performed in duplicate, and repetition was performed when the coefficient of variation was greater than 5.0%.

**Determination of fatty acid profile.** The fatty acid profile of the foods was determined by gas chromatography, according to ISO [6], at the Laboratory of Biochemistry of Microorganisms and Plants, Department of Technology at the School of Agrarian and Veterinary Sciences of the São Paulo State University "Júlio de Mesquita Filho" (Jaboticabal, SP, Brazil).

**Determination of minerals.** Samples were prepared by a wet method prior to mineral analyses. A 200mg sample was conditioned in a 100mL test with 4mL of concentrated nitric acid ($HNO_3$) solution. After this process, the tubes rested for 30 minutes. Next, samples were placed on a heating plate and boiled until the volume of nitric acid was reduced by about half. Then, tubes were cooled and 1mL of perchloric acid ($HClO_4$) was added to each sample, which was reheated until the sample volume reached 2mL, and cooled again.

After sample preparation, calcium (Ca), phosphorus (P), potassium (K), sodium (Na), magnesium (Mg), iron (Fe), manganese (Mn), copper (Cu) and zinc (Zn) were determined by inductively coupled plasma optical emission spectrometry (ICP-OES; ICPE-9000, Shimadzu do Brasil, Barueri, SP, Brazil). The operational conditions are presented in Table 2. In order to ensure the accuracy of the results, Milli-Q water was used to flush the system between each analysis, and the system was cleaned with a 1.0% nitric acid solution after every five samples analyzed.

**Table 2. Operational conditions of inductively coupled plasma optical emission spectrometry (ICP-OES) with axial configuration.**

| Parameter | Technical features |
|---|---|
| Radio frequency power (W) | 1200 |
| Vaporized plasma flow rate (L/min) | 10 |
| Auxiliary gas flow rate (L/min) | 0.6 |
| Exposure time (s) | 30 |
| Nebulizer gas flow rate (L/min) | 0.7 |
| Nebulizer type | Concentric |
| Spray chamber | Cyclone |
| Number of replicates | 3 |

The calibration curves were prepared through standardized multi-element solutions of 100mg/L for the elements Ca, Cl, Cu, Fe, K, Mg, Mn, Na, P, and Zn. Calibration curves were prepared from 0.1 to 5mg/L for Cu, Zn, Fe, and Mn, and from 0.5 to 100mg/L for Ca, P, Mg, Na and K.

Chloride (Cl) and iodine (I) were not measured because the digestion methodology employed does not allow it due to these elements' high ionization energy. For example, in iodine, only 29.0% of atoms are ionized in argon plasma [7]. Selenium was not measured because it would require a hydride generator coupled to ICP-OES, which was not available to us. The preparation of the wet solution samples, as well as the ICP-OES analyses, were carried out at the Multi-user Laboratory of Animal Nutrition and Bromatology of the Department of Animal Nutrition and Production at the School of Veterinary Medicine and Animal Science at the University of São Paulo (Pirassununga, SP, Brazil).

**Determination of the amino acid profile.** The amino acid profile was determined by high performance liquid chromatography (HPLC) according to Spitze et al. [8] at the Monogastric Nutrition Laboratory of the Department of Animal Nutrition and Production at the School of Veterinary Medicine and Animal Science at the University of São Paulo (Pirassununga, SP, Brazil).

The results were reported in unit per 100 grams of dry matter (DM). It was also calculated how much of each nutrient each food provides in units per kg of metabolic weight for animals with low energy requirement, based on $80kcal/kg^{0.75}$ and $52kcal/kg^{0.67}$ of MER for dogs and cats, respectively [9]. Although FEDIAF (2019) [9] recommendations in units per 100g of DM takes into account only the equations $95kcal/kg^{0.75}$ and $110kcal/kg^{0.75}$ for dogs, and $75kcal/kg^{0.67}$ and $100kcal/kg^{0.67}$ for cats, FEDIAF (2019) [9] also reports that there are animals which can present lower metabolizable energy requirement (MER). Therefore, we chose to use the equations for low energy requirement to ensure that all these animals would be receiving at least the minimum of each essential nutrient per day. The metabolizable energy of the foods were estimated by the most current method described in the NRC [10].

## Statistical analysis

The results obtained were compared with the FEDIAF (2019) [9] and AAFCO (2019) [11] recommendations in unit per 100 grams of food and with the FEDIAF (2019) [9] in unit per kg of metabolic weight.

## Results

The results of the laboratory analyses of the vegan foods for dogs, the FEDIAF [9] and AAFCO [11] recommendations in unit per 100 grams of dry matter are described in Table 3. All foods analyzed met the recommendations of the FEDIAF [9] and AAFCO [11] for CP and fat.

One vegan dog food (Food A) had a methionine and calcium deficiency according to the minimum recommendations by FEDIAF [9] for adult dog maintenance. All evaluated dog foods had Ca:P ratios below the minimum recommendations by FEDIAF [9] and AAFCO [11].

In regard to Cu, all foods evaluated exceeded the maximum safe recommendations for inclusion of this mineral in dog foods based on FEDIAF [9]. These maximum recommendations are based on the legal limits of the European Union in compliance with the Regulation 1831/2003/EC. Based on these same recommendations, one of the foods (Food A) also had excess Zn, and Food C presented Fe excess.

The nutrient analysis of the only commercial vegan food available on the Brazilian market formulated for cats and the recommended nutrient levels for adult cat maintenance by the FEDIAF [9] and AAFCO [11] in unit per 100 grams of dry matter are described in Table 4.

The cat food evaluated did not contain any arachidonic acid in its composition, which is an essential fatty acid for this species and has a minimum recommendation for cats by FEDIAF [9] and AAFCO [11].

The vegan cat food also presented values of calcium and calcium:phosphorus ratio below the minimum recommended by FEDIAF [9]. Potassium was also lower than the minimum recommendations of FEDIAF [9] and AAFCO [11]. Furthermore, when taking into consideration the maximum recommendations, based on the legal limits of the European Union in compliance with regulation 1831/2003/EC, the vegan food for cats presented copper and zinc levels above recommendation.

The results expressed in unit per kg of metabolic weight of the vegan dog foods are described in Table 5, and those of the vegan cat food are presented in Table 6.

The vegan cat diet only supplied 75.4% of the minimum protein required for a cat with an energy requirement of 52 kcal/kg$^{0.67}$ (Table 6). Arginine deficiency was also observed, for which the cat food supplied only 59.0% of the minimum recommendations for a vegan cat with an energy requirement of 52 kcal/kg$^{0.67}$. Besides these deficiencies, the amino sulphonic acid taurine was also below the minimum recommendation. Regarding the nutrient recommendations per kg of metabolic weight for animals with lower MER, the same deficiencies were observed when comparing to results in unit per 100 grams of dry matter.

## Discussion

Veganism can be considered a strict form of vegetarianism, where an individual is not allowed to eat any food products or derivatives of animal origin. The humanization of pets has led around ¼ of vegan pet owners to seek a similar diet for their animals in the US [1], leading to some growth in the vegan market. However, it is challenging to formulate a complete and balanced diet for facultative or obligate carnivores based on plants. In the present study, we can highlight that commercial products available for dogs and cats that claim to be vegan may have some inconsistency. In the ingredient composition statement on the label, food C claims to contain vitamin D$_3$. Vitamin D is presented in two forms in nature: cholecalciferol (vitamin D$_3$), found in animal products and tissues, and ergocalciferol (vitamin D$_2$), found in products of plant origin [12]. Thus, the presence of vitamin D$_3$ indicates animal sourcing, which should not be present in a vegan product by definition. This demonstrates the contestable truth of the evaluated product.

Laboratory analyses of the four vegan foods showed that none of them met all of the nutritional recommendations of FEDIAF [9] and AAFCO (2019) [11] for the species for which they were intended. Levels below the minimum recommendation for various nutrients were observed, including amino acids such as arginine and methionine, and minerals such as

**Table 3. Nutrient analysis of three commercial vegan diets for dogs and recommended nutrient levels for adult dog maintenance, based on FEDIAF (2019) [maintenance energy requirement (MER) of 95 kcal of metabolizable energy (ME)/kg$^{0.75}$] and AAFCO (2019)—unit per 100 g dry matter.**

| Nutrient | Foods | | | FEDIAF (2019) | | AAFCO (2019) | |
|---|---|---|---|---|---|---|---|
| | Food A | Food B | Food C | Minimum | Maximum | Minimum | Maximum |
| **Protein** | | | | | | | |
| **Crude protein (g)** | 24.36 | 27.96 | 30.04 | 21.00 | NA | 18.00 | NA |
| **Arginine (g)** | 0.77 | 1.06 | 0.97 | 0.60 | NA | 0.51 | NA |
| **Cystine (g)** | 0.81 | 0.95 | 0.90 | NA | NA | NA | NA |
| **Histidine (g)** | 0.54 | 0.71 | 0.66 | 0.27 | NA | 0.19 | NA |
| **Isoleucine (g)** | 0.96 | 1.18 | 1.24 | 0.53 | NA | 0.38 | NA |
| **Leucine (g)** | 1.53 | 2.76 | 3.46 | 0.95 | NA | 0.68 | NA |
| **Lysine (g)** | 1.05 | 1.35 | 1.47 | 0.46 | NA | 0.63 | NA |
| **Methionine (g)** | 0.41* | 0.71 | 0.52 | 0.46 | NA | 0.33 | NA |
| **Methionine + cystine (g)** | 1.22 | 1.66 | 1.42 | 0.88 | NA | 0.65 | NA |
| **Phenylalanine (g)** | 1.19 | 1.58 | 1.74 | 0.63 | NA | 0.45 | NA |
| **Phenylalanine + tyrosine (g)** | 1.99 | 2.73 | 3.02 | 1.03 | NA | 0.74 | NA |
| **Taurine (g)** | 0.14 | 0.16 | 0.15 | NA | NA | NA | NA |
| **Threonine (g)** | 0.90 | 1.00 | 1.00 | 0.60 | NA | 0.48 | NA |
| **Tryptophan (g)** | NP | NP | NP | 0.20 | NA | 0.16 | NA |
| **Tyrosine (g)** | 0.80 | 1.14 | 1.28 | NA | NA | NA | NA |
| **Valine (g)** | 1.01 | 1.19 | 1.27 | 0.68 | NA | 0.49 | NA |
| ***Fat*** | | | | | | | |
| **Crude fat (g)** | 13.43 | 13.96 | 8.08 | 5.50 | NA | 5.50 | NA |
| **Linoleic acid (ω-6) (g)** | 6.57 | 6.91 | 3.98 | 1.53 | NA | 1.1 | NA |
| **Arachidonic acid (ω-6) (mg)** | 0 | 0 | 0 | NA | NA | NA | NA |
| **Alpha-linolenic acid (ω-3) (g)** | 0.93 | 1.09 | 0.41 | NA | NA | NA | NA |
| ***Minerals*** | | | | | | | |
| **Ash (g)** | 4.91 | 8.03 | 8.31 | NA | NA | NA | NA |
| **Calcium (g)** | 0.56* | 0.86 | 0.99 | 0.58 | 2.50 | 0.50 | 2.50 |
| **Phosphorus (g)** | 0.93 | 1.03 | 1.03 | 0.46 | 1.60 | 0.40 | 1.60 |
| **Ca:P ratio (g)** | 0.60** | 0.83** | 0.96** | 1 | 2 | 1 | 2 |
| **Potassium (g)** | 0.65 | 1.27 | 0.40 | 0.58 | NA | 0.60 | NA |
| **Sodium (g)** | 0.07 | 0.40 | 0.35 | 0.12 | NA | 0.08 | NA |
| **Chloride (g)** | NP | NP | NP | 0.17 | NA | 0.12 | NA |
| **Magnesium (g)** | 0.19 | 0.22 | 0.11 | 0.08 | NA | 0.06 | NA |
| **Copper (mg)** | 3.00*** | 6.10*** | 5.30*** | 0.83 | 2.80 | 0.73 | NA |
| **Iodine (mg)** | NP | NP | NP | 0.12 | 1.10 | 0.10 | 1.10 |
| **Iron (mg)** | 39.00 | 40.00 | 75.00*** | 4.17 | 68.18 | 4.00 | NA |
| **Manganese (mg)** | 7.20 | 5.30 | 9.60 | 0.67 | 17.00 | 0.50 | NA |
| **Selenium (µg)** | NP | NP | NP | 35.00 | 56.8 | 35.00 | 200.00 |
| **Zinc (mg)** | 24.00*** | 18.00 | 21.00 | 8.34 | 22.70 | 8.00 | NA |
| ***Crude fiber*** | 6.29 | 8.07 | 4.29 | NA | NA | NA | NA |

* Less than the minimum recommended by FEDIAF (2019).

** Less than the minimum recommended by FEDIAF (2019) and AAFCO (2019).

*** Higher than the legal limit of FEDIAF (2019).

NA, not applicable because a minimum or maximum value has not been established for these nutrients; NP, not performed.

calcium, potassium and sodium. The main findings of this study were levels below the detection limit for arachidonic acid (essential for cats), as well as failure to comply with the minimum recommendation for the Ca:P ratio in all analyzed foods.

**Table 4. Nutrient analysis of a commercial vegan food for cats and recommended nutrient levels for maintenance adult cat based on FEDIAF (2019) [maintenance energy requirement (MER) of 75 kcal metabolizable energy (ME)/kg$^{0.67}$] and AAFCO (2019)—unit per 100 g dry matter.**

| Nutrient | Food D | FEDIAF (2019) | | AAFCO (2019) | |
|---|---|---|---|---|---|
| | | Minimum | Maximum | Minimum | Maximum |
| **Protein** | | | | | |
| **Crude protein (g)** | 36.30 | 33.30 | NA | 26.00 | NA |
| **Arginine (g)** | 1.23* | 1.30 | NA | 1.04 | NA |
| **Cystine (g)** | 1.30 | NA | NA | NA | NA |
| **Histidine (g)** | 0.85 | 0.35 | NA | 0.31 | NA |
| **Isoleucine (g)** | 1.48 | 0.57 | NA | 0.52 | NA |
| **Leucine (g)** | 4.05 | 1.36 | NA | 1.24 | NA |
| **Lysine (g)** | 1.65 | 0.45 | NA | 0.83 | NA |
| **Methionine (g)** | 1.03 | 0.23 | NA | 0.20 | 1.15 |
| **Methionine + cystine (g)** | 2.33 | 0.45 | NA | 0.40 | NA |
| **Phenylalanine (g)** | 2.12 | 0.53 | NA | 0.42 | NA |
| **Phenylalanine + tyrosine (g)** | 3.69 | 2.04 | NA | 1.53 | NA |
| **Taurine (g)** | 0.17 | 0.13 | NA | 0.10 | NA |
| **Threonine (g)** | 1.17 | 0.69 | NA | 0.73 | NA |
| **Tryptophan (g)** | NP | 0.17 | NA | 0.16 | 1.17 |
| **Tyrosine (g)** | 1.57 | NA | NA | NA | NA |
| **Valine (g)** | 1.51 | 0.68 | NA | 0.62 | NA |
| *Fat* | | | | | |
| **Crude fat** | 17.67 | 9.00 | NA | 9.00 | NA |
| **Linoleic acid (ω-6)** | 8.58 | 0.67 | NA | 0.60 | NA |
| **Arachidonic acid (ω-6) (mg)** | 0** | 8.00 | NA | 20.00 | NA |
| **Alpha-linolenic acid (ω-3)** | 1.18 | NA | NA | NA | NA |
| *Minerals* | | | | | |
| **Mineral matter (g)** | 6.45 | NA | NA | NA | NA |
| **Calcium (g)** | 0.72* | 0.79 | NA | 0.60 | NA |
| **Phosphorus (g)** | 1.07 | 0.67 | NA | 0.50 | NA |
| **Ca / P ratio (g)** | 0.67* | 1.00 | 2.00 | NA | NA |
| **Potassium (g)** | 0.54** | 0.80 | NA | 0.60 | NA |
| **Sodium (g)** | 0.20 | 0.10 | NA | 0.20 | NA |
| **Chloride (g)** | NP | 0.15 | NA | 0.30 | NA |
| **Magnesium (g)** | 0.17 | 0.05 | NA | 0.04 | NA |
| **Copper (mg)** | 7.60*** | 0.67 | 2.80 | 0.50 | NA |
| **Iodine (mg)** | NP | 0.17 | 1.10 | 0.06 | 0.90 |
| **Iron (mg)** | 39.00 | 10.70 | 68.18 | 8.00 | NA |
| **Manganese (mg)** | 7.00 | 0.67 | 17.00 | 0.76 | NA |
| **Selenium (μg)** | NP | 40.00 | 56.80 | 30.00 | NA |
| **Zinc (mg)** | 24.00*** | 10.00 | 22.70 | 7.50 | NA |
| *Crude fiber* | 8.97 | NA | NA | NA | NA |

\* Less than the minimum recommended by FEDIAF (2019).

\*\* Less than the minimum recommended by FEDIAF (2019) and AAFCO (2019).

\*\*\* Higher than legal limit of FEDIAF (2019).

NA, not applicable because a minimum or maximum value has not been established for these nutrients; NP, not performed.

Arachidonic acid (AA) is an essential fatty acid for cats because these animals have a low delta-6-desaturase enzyme activity, which is responsible for the conversion of linoleic acid into

**Table 5. Amount of nutrients per kg$^{0.75}$ provided by vegan dog foods with an energy requirement of 80kcal/kg$^{0.75}$ and minimum recommended nutrient levels per kg$^{0.75}$ (FEDIAF, 2019).**

| Nutrients | Nutrient levels per kg$^{0.75}$ | | | Minimum recommended nutrient levels per kg$^{0.75}$ [10] |
|---|---|---|---|---|
| | Food A | Food B | Food C | |
| **Protein (g)** | 5.04 | 6.11 | 6.30 | 4.95 |
| **Arginine (g)** | 0.16 | 0.23 | 0.20 | 0.14 |
| **Histidine (g)** | 0.11 | 0.16 | 0.14 | 0.06 |
| **Isoleucine (g)** | 0.20 | 0.26 | 0.26 | 0.13 |
| **Leucine (g)** | 0.32 | 0.60 | 0.73 | 0.23 |
| **Lysine (g)** | 0.22 | 0.30 | 0.31 | 0.12 |
| **Methionine (g)** | 0.08* | 0.16 | 0.11 | 0.11 |
| **Methionine + cystine (g)** | 0.25 | 0.36 | 0.30 | 0.21 |
| **Phenylalanine (g)** | 0.25 | 0.35 | 0.37 | 0.15 |
| **Phenylalanine + tyrosine (g)** | 0.41 | 0.60 | 0.63 | 0.24 |
| **Threonine (g)** | 0.19 | 0.22 | 0.21 | 0.14 |
| **Tryptophan (g)** | NP | NP | NP | 0.05 |
| **Valine (g)** | 0.21 | 0.26 | 0.27 | 0.16 |
| **Fat (g)** | 2.79 | 3.05 | 1.70 | 1.51 |
| **Linoleic acid (g)** | 1.36 | 1.53 | 0.82 | 0.36 |
| **Minerals** | | | | |
| **Calcium (g)** | 0.12* | 0.19 | 0.21 | 0.14 |
| **Phosphorus (g)** | 0.19 | 0.23 | 0.22 | 0.11 |
| **Potassium (g)** | 0.13* | 0.28 | 0.08* | 0.14 |
| **Sodium (g)** | 0.01* | 0.09 | 0.07 | 0.03 |
| **Chloride (g)** | NP | NP | NP | 0.04 |
| **Magnesium (g)** | 0.04 | 0.05 | 0.02 | 0.02 |
| **Trace elements** | | | | |
| **Copper (mg)** | 0.63 | 1.34 | 1.11 | 0.20 |
| **Iodine (mg)** | NP | NP | NP | 0.03 |
| **Iron (mg)** | 8.10 | 8.77 | 15.76 | 1.0 |
| **Manganese (mg)** | 1.51 | 1.17 | 2.02 | 0.16 |
| **Selenium (µg)** | NP | NP | NP | 8.25 |
| **Zinc (mg)** | 4.99 | 3.86 | 4.40 | 2.0 |

NP, not performed.

* Less than the minimum recommended by FEDIAF (2019).

AA. Fats of animal origin are good sources of AA, whereas vegetable oils are absent. The study of lipid metabolism in felines instigated the interest of the scientific community in the 1970s, when Rivers et al. [13] observed while studying liver samples that cats had very low activity of delta-6-dessaturase enzyme. After this discovery, effects of low AA diets for cats were studied and it was shown that AA deficiency in cats can result in thrombocytopenia, impairment of platelet aggregation and inability to conceive [14]. Pawlosky and Salem [15] fed AA-free diets supplemented with 1% corn oil to cats before mating and throughout gestation. All females cycled and mated, but a high incidence of congenital defects and low litter viability were observed. The vegan foods analyzed in our study used soybean oil as the only source of fat, so it was expected that AA would not be detected in these products. Some marine algae such as *Ascophyllum nodosum* may contain AA and could, therefore, be used as a source of this fatty acid. On the label of the commercial vegan cat food analyzed, it was stated that there was an

**Table 6. Amount of nutrients per $kg^{0.67}$ provided by a vegan cat food with an energy requirement of 52 $kcal/kg^{0.67}$ and minimum recommended nutrient levels per $kg^{0.67}$ (FEDIAF, 2019).**

| Nutrients | Nutrient levels per $kg^{0.67}$ | Minimum recommended nutrient levels per $kg^{0.67}$ [10] |
|---|---|---|
| | Food D | |
| Protein (g) | 4.71* | 6.25 |
| Arginine (g) | 0.16* | 0.25 |
| Histidine (g) | 0.11 | 0.08 |
| Isoleucine (g) | 0.19 | 0.12 |
| Leucine (g) | 0.53 | 0.29 |
| Lysine (g) | 0.21 | 0.09 |
| Methionine (g) | 0.13 | 0.04 |
| Methionine + cystine (g) | 0.3 | 0.09 |
| Phenylalanine (g) | 0.27 | 0.12 |
| Phenylalanine + tyrosine (g) | 0.48 | 0.44 |
| Taurine (g) | 0.02* | 0.15 |
| Threonine (g) | 0.15 | 0.04 |
| Tryptophan (g) | NP | 0.15 |
| Valine (g) | 0.2 | 0.03 |
| Fat | 2.29 | 2.25 |
| Linoleic acid (g) | 1.11 | 0.13 |
| Arachidonic acid (mg) | ND* | 1.50 |
| Minerals | | |
| Calcium (g) | 0.09* | 0.15 |
| Phosphorus (g) | 0.14 | 0.13 |
| Potassium (g) | 0.07* | 0.15 |
| Sodium (g) | 0.02 | 0.02 |
| Chloride (g) | NP | 0.03 |
| Magnesium (g) | 0.02 | 0.01 |
| Trace elements | | |
| Copper (mg) | 0.98 | 0.13 |
| Iodine (mg) | NP | 0.03 |
| Iron (mg) | 5.07 | 2.00 |
| Manganese (mg) | 0.91 | 0.13 |
| Selenium (µg) | NP | 7.50 |
| Zinc (mg) | 3.11 | 1.88 |

* Less than the minimum recommended by FEDIAF (2019).

NP, not performed; ND, not detected.

inclusion of algae meal, although the information about which species of algae and the amount added were not included. Unlike the US, Brazilian laws do not require that ingredients are added in order of highest to lowest percent inclusion.

All foods presented a Ca:P ratio below the minimum (1:1) recommended by FEDIAF [9]. Regarding Ca, two foods (one for dogs and one for cats) presented concentrations below recommendations. A similar result was found by Gray et al. [16], who analyzed two commercial vegan foods for cats and one of them presented calcium concentrations well below the minimum recommended for the species. In general, by-products of animal origin have higher calcium concentrations when compared to products of plant origin [10], which may explain the results found. However, according to the labels of all diets, there was an inclusion of calcium

supplementation in the form of calcium carbonate and dicalcium phosphate. These were not sufficient to achieve above the minimum recommendation. In humans, it has been shown that vegan people consume less calcium [17] and present lower bone mineral density [18,19] compared to non-vegans, which may increase the risk of bone fracture [20].

The Ca:P ratio below the minimum recommendation and the low concentrations of Ca in the diet may result in secondary nutritional hyperparathyroidism (SNH) in dogs and cats [21, 22]. This abnormality was quite common in the past because of unbalanced diets provided to pets [22, 23, 24, 25, 26]. However, SNH is uncommon nowadays due to the large use of complete and balanced commercial foods that supply the animal's calcium requirements. The SNH may result in rubber jaw syndrome, a group of lesions characterized by the loss of the lamina dura of the teeth. Symptoms include loose teeth, osteopenia of the skull bones, and a rubbery aspect of the jaw in consequence of the replacement of the mineral structure by fibrous tissue [26]. Furthermore, hypocalcemia is usually present in this syndrome, which can cause muscle spasms, convulsion, and osteopenia, which causes bone fractures [22].

One of the dog foods did not meet the minimum recommendation for the amino acid methionine. This result was expected since methionine occurs at low levels in vegetable protein based diets [10]. The product label of this food stated that DL-methionine was supplemented, and yet its concentrations were below the minimum recommendation. Currently, there is a concern regarding methionine for dogs since the low consumption of this amino acid may be indirectly related to an increased risk of developing dilated cardiomyopathy (DCM) [27]. Methionine is a precursor of taurine, an important amino acid for myocardial function [28, 29]. Dogs are able to synthesize taurine from sulfur amino acids such as methionine and cysteine, mainly in the liver and central nervous system, through the transsulfuration pathway [28]. The relationship between taurine deficiency and DCM in dogs was first studied by Kramer et al. [30], who observed that a subset of dogs with DCM had taurine deficiency. Further studies have shown that taurine supplementation for dogs with DCM associated with the deficiency of this amino acid resulted in resolution of myocardial changes and allowed the removal of support medication for most of these dogs [29, 31, 32, 33]. This data suggest a direct link between taurine deficiency and DCM development. Fascetti et al. [29] assessed three taurine-deficient Golden Retrievers that were fed a vegetarian diet formulated by the owner, and according to the authors this deficiency may have occurred due to possible low dietary sulfur amino acids precursors of taurine.

Although taurine deficiency may occur as a consequence of dietary deficiency of methionine or cysteine, there are other nutritional factors that may be involved, such as decreased bioavailability of these sulfur amino acids, abnormal enterohepatic bile acid recycling, altered taurine intestinal metabolism and urinary loss of taurine [27, 34]. In the present study, the vegan food with low methionine concentration may represent a risk factor for the development of DCM. In addition, this food presented a high inclusion of soybean meal in its composition, which was the main protein ingredient used in its formulation. It has been observed in cats that soy protein decreases plasma taurine concentrations as a consequence of increased bile acid loss that is conjugated to taurine through microbial degradation and acceleration of cholecystokinin-mediated bile acid turnover [35]. Moreover, methionine is the limiting amino acid of most legumes like soybean. Thus, the high inclusion of this ingredient may also possibly contribute to the development of DCM.

The analyzed cat food did not meet the minimum recommendation for arginine. This result corroborates a previous study by Gray et al. [16]. The low arginine was expected because this amino acid is usually found in small amounts in plant-based diets, and according to the product label the food was not supplemented with arginine. Cats are more sensitive to arginine deficiency than dogs [9, 36]. Arginine acts as a precursor of ornithine, which is an intermediate

in the urea cycle. This allows large quantities of ammonia from a high-protein meal to be converted to urea before excretion [37]. Lack of arginine in cats may lead to hyperammonemia and severe uremia, causing symptoms such as emesis, ataxia, muscle spasms, and hyperesthesia. [38]. The higher sensitivity of cats to arginine-deficient diets may be explained by two reasons. First, most animals are able to use glutamate and proline as precursors for the synthesis of ornithine in the intestinal mucosa, but cats are unable to utilize these amino acids as precursors because they have low pyrroline-5-carboxylate synthetase enzyme activity in their intestinal mucosa [39]. In addition, cats present low activity of another essential enzyme for this process, the ornithine aminotransferase [37]. Furthermore, felines are unable to synthesize arginine from ornithine, even if they consume a diet with high amounts of ornithine. It occurs due to the fact that cats are unable to synthesize citrulline in the intestinal mucosa, a molecule that acts as a precursor in the synthesis of arginine from ornithine. Although they produce citrulline in the liver, this molecule is unable to exit the hepatocytes and be transported to the kidneys, where conversion to arginine could potentially occur [40].

In addition to the nutritional recommendation based on DM, FEDIAF [9] also has recommendations on a per kg of metabolic weight for dogs and cats. This is important since recommendations based on DM for cats take into account a MER of $75kcal/kg^{0.67}$ and $100kcal/kg^{0.67}$, but some cats may present lower MER due to a sedentary lifestyle, neutering/spaying or sometimes pathological conditions that lead to hyporexia. Hence, FEDIAF [9] suggests that MER of inactive adult cats can also range from 52 to 75kcal per kg of metabolic weight. Loureiro et al. [41] reported a mean MER for adult cats of $59.5\pm2.2$ $Kcal/BW^{0.67}$/day. For dogs, this comparison is also important, since nutritional recommendations based on DM take into account the energy requirement of $95kcal/kg^{0.75}$ and $110kcal/kg^{0.75}$, but there are animals with lower energy requirements. According to the recommendations by FEDIAF [9], the MER of inactive dogs can vary from 80 to $120kcal/kg^{0.75}$. Pedrinelli et al. [42] observed mean MER for adult dogs of $86.1\pm19.8$ $kcal/BW^{0.75}$. For those animals with low MER, the nutrient concentrations in the food should be higher, to ensure the intake of at least the minimum requirements. Therefore, in the present study a comparison was made among foods' nutritional composition and recommendations per kg of metabolic weight for cats with MER of $52kcal/kg^{0.67}$, and dogs with MER of $80kcal/kg^{0.75}$, as mentioned by FEDIAF [9].

The results on a per kg of metabolic weight basis showed values of protein and taurine below the minimum recommendation in the cat food, in addition to the same deficiencies found in the results per unit of DM. This data corroborates with a study by Loureiro et al. [41], in which they observed that a commercial food with 28% CP on a DM basis did not meet the minimum recommendation of protein intake in $g/kg^{0.67}$, although it did support the requirement for CP based on DM. This happened due to the low amount of food ingested by these animals to maintain weight, which had an average MER of 59.5 $kcal/kg^{0.67}$. In that study, cats that consumed the protein deficient diet during a 60-day period lost 90 grams of lean body mass. It was also observed that the protein intake required to maintain stable lean body mass was $5.17g/kg^{0.67}$, higher than that provided by the vegan food analyzed in our study for cats. Based on the work by Loureiro et al. [41], it is possible that the intake of the vegan cat food evaluated in the present study may lead to lean body mass loss in cats with low energy requirements, despite presenting 36.3% of CP on a DM basis.

Regarding taurine consumption below the recommendation per $kg^{0.67}$, two previous studies have also observed a deficiency of this amino sulphonic acid in commercial vegetarian and vegan cat foods [16, 43]. This is a serious implication when feeding cats a plant-based diet. In the work by Wakefield et al. [44], they measured blood taurine concentrations of cats that were fed vegetarian diets and observed that, of the 17 cats evaluated, three had taurine concentrations below the reference range.

Taurine is an amino sulphonic acid that is only found in animal products. In cat foods, even with the inclusion of animal ingredients, taurine supplementation is usually required to achieve the minimum requirement for this amino acid. In the case of diets without animal products, taurine supplementation becomes mandatory. The vegan cat food analyzed in this study had taurine supplementation, however, the concentration found was below the recommendation, based on unit per $kg^{0.67}$ for cats with MER of $52kcal/kg^{0.67}$.

Taurine plays an essential role in the digestive system of cats. This species conjugates bile acids exclusively with taurine, unlike dogs that may occasionally conjugate bile salts with glycine [16, 45]. The intestinal lumen microbial degradation increases the loss of taurine in cats [46]. In addition, cats show limited hepatic production of taurine from sulfur amino acids, such as methionine and cysteine, which results from the low activity of the enzyme cysteine dioxygenase. This low enzymatic activity limits the production of cysteine sulfate, an intermediate in taurine synthesis. Felines also have limited activity of cysteine sulfinic acid decarboxylase, an enzyme responsible for converting cysteine sulfate to taurine. Additionally, cats have a high activity of aspartate aminotransferase, which transforms cysteine sulfate into pyruvate, rather than allowing it to be decarboxylated to hypotaurine, and therefore into taurine [46, 47, 48]. All these associated factors result in a dietary requirement for taurine in felines.

It is important to emphasize that the cat vegan food analyzed may also be a risk factor for taurine deficiency due to its high inclusion of soybean meal. It is known that soy protein may decrease plasma concentrations of taurine in cats when compared to casein, which seems to be related to increased use of taurine for conjugation of bile acids, increased intestinal loss of taurine by microbial degradation and accelerated cholecystokinin-mediated turnover of bile acids [35, 49, 50]. Therefore, caution with supplementation of taurine in vegan foods with high inclusion of soy protein should be even greater.

In cats, dietary deficiency of taurine may lead to cardiac abnormalities such as dilated cardiomyopathy, since taurine is related to myocardial function and it is essential for the maintenance of cardiac contractility [51, 52]. Novotny et al. [53] fed a taurine deficient diet to 23 cats over a period of 6 to 15 months and most of the animals presented decreased myocardial function assessed by echocardiography.

Taurine also plays an important role in maintaining the structural integrity of the retina. Dietary deficiency of this amino sulphonic acid can lead to photoreceptor cell degeneration [54, 55, 56, 57, 58]. In a study carried out by Jacobson et al. [58], eight male cats were fed a taurine deficient diet for 5 to 9 months and all animals developed abnormalities in the retina detected by fundoscopy and electroretinography. In addition, taurine deficiency can impair the reproductive performance of cats with consequences on kitten development [59].

In the present study, nutrient digestibility and bioavailability were not assessed. However, it is generally recognized that plant protein sources have lower digestibility compared to animal protein [43, 60]. Funaba et al. [61] observed in adult cats that a diet in which meat meal was used as a protein source resulted in higher apparent nitrogen absorption and retention, and higher dry matter digestibility when compared to a corn gluten meal based diet. In another study conducted by Bednar et al. [62], when comparing the digestibility of different protein sources by adult dogs, total tract crude protein digestibility of poultry meal was greater than soybean meal, poultry by-product meal, and beef and bone meal (87.5% vs average 82.2%). These authors also observed lower organic matter digestibility for the soybean meal treatment. Hill et al. [63] observed that the partial replacement of beef with textured soy protein (100% beef vs 57% soy textured protein and 43% beef) in canned foods fed to cannulated dogs had a decrease in the total tract apparent digestibility of crude protein, energy, dry matter, sodium, potassium and iron. These authors also observed lower pre-cecal apparent digestibility of essential amino acids arginine, histidine, isoleucine, leucine, valine, lysine, phenylalanine and

threonine (tryptophan and methionine were not assessed) when dogs were fed the textured soy protein treatment. These data suggest that even if vegan food has nutrient concentrations above the minimum recommendations of FEDIAF and AAFCO, there may still be a risk of nutritional deficiency due to the lower digestibility and bioavailability of vegetable-based nutrients.

Vitamin analyses were not performed in the present study. However, the label of the vegan cat food (Food D) reported vitamin D inclusion in the form of vitamin $D_2$ (ergocalciferol), which is present in vegetables. It is known that vitamin $D_2$ in cats is not as efficient at maintaining 25(OH)D serum concentrations as vitamin $D_3$ (cholecalciferol) [64]. According to Morris [64], the efficiency of use of ergocalciferol for synthesis of 25(OH)D is 0.7 that of cholecalciferol in the cat. The 25(OH)D is an intermediate metabolite of vitamin D. It is synthesized in the liver and converted to 1,25(OH)$_2$D in the proximal renal tubules [65]. 1,25(OH)$_2$D is the active metabolite that performs classical hormonal functions related to bone metabolism and calcium and phosphorus homeostasis in the body, in addition to non-classical functions such as immune-modulating activity [66,67,68], protection against autoimmune reactions [69, 70], antitumoral activity [71,72,73,74] and anti-inflammatory activity [75,76]. Because of this peculiarity of cats in relation to vitamin D metabolism, if the analyzed vegan food included vitamin $D_2$ in their formula, this may likely result in long-term vitamin D deficiency.

Concentrations of copper and zinc higher than legal limits of FEDIAF were found in the diets evaluated, and one vegan dog food had iron concentrations above the legal limit of FEDIAF. There is insufficient information in the literature about the safe upper limit or nutritional limit for copper, zinc and iron intake for dogs and cats, so it is not known if levels observed in these foods may imply health risks for dogs and cats. However, transition metals like copper are known to cause oxidative damage when in excess [77], which can potentially lead to liver cell damage or death, and abnormalities associated with oxidative stress.

The limitations of the present study were that some essential nutrients were not quantified in the vegan diets, such as water-soluble and fat-soluble vitamins, as well as minerals iodine, chloride and selenium.

In conclusion, all foods tested had one or more nutrients lower than what is recommended, while some presented excess zinc and copper. Although these foods are commercially available in the Brazilian market, they should not be recommended for dogs and cats, because dietary inadequacies observed may lead to health risks and even death. Furthermore, manufacturers should review their formulations to ensure the nutritional adequacy of these foods.

## Acknowledgments

We would like to thank Renata Maria Consentino Conti and Simi Luisa Durante Aflalo for technical assistance.

## Author Contributions

**Conceptualization:** Rafael Vessecchi Amorim Zafalon, Thiago Henrique Annibale Vendramini, Marcio Antonio Brunetto.

**Data curation:** Rafael Vessecchi Amorim Zafalon, Thiago Henrique Annibale Vendramini.

**Formal analysis:** Rafael Vessecchi Amorim Zafalon, Thiago Henrique Annibale Vendramini, Vivian Pedrinelli, Fabio Alves Teixeira, Mariana Fragoso Rentas.

**Funding acquisition:** Fabio Alves Teixeira.

**Investigation:** Roberta Bueno Ayres Rodrigues, Vivian Pedrinelli.

**Methodology:** Roberta Bueno Ayres Rodrigues, Vivian Pedrinelli, Mariana Fragoso Rentas.

**Project administration:** Marcio Antonio Brunetto.

**Software:** Mariana Fragoso Rentas.

**Visualization:** Larissa Wünsche Risolia.

**Writing – original draft:** Larissa Wünsche Risolia, Roberta Bueno Ayres Rodrigues.

**Writing – review & editing:** Mariana Pamplona Perini, Isabella Corsato Alvarenga.

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
