## [Decision Letter · Decision Letter 0]

7 Nov 2019

PONE-D-19-28038

Nutritional inadequacies in commercial vegan foods for dogs and cats

PLOS ONE

Dear Dr. Brunetto,

Thank you for submitting your manuscript to PLOS ONE. After careful consideration, we feel that it has merit but does not fully meet PLOS ONE’s publication criteria as it currently stands. Therefore, we invite you to submit a revised version of the manuscript that addresses the points raised during the review process.

Please take into account both reviewers' comments and carefully address all their queries. The information provided by your manuscript is useful and, in my opinion, worth to be published, but the ms needs some reworking (including language and grammar - in this sense Rev. 2 provided some suggestions and edits on the attached PDF).

We would appreciate receiving your revised manuscript by Dec 22 2019 11:59PM. To enhance the reproducibility of your results, we recommend that if applicable you deposit your laboratory protocols in protocols.io, where a protocol can be assigned its own identifier (DOI) such that it can be cited independently in the future. For instructions see: http://journals.plos.org/plosone/s/submission-guidelines#loc-laboratory-protocols

We look forward to receiving your revised manuscript.

Kind regards,

Nicoletta Righini, PhD

Academic Editor

PLOS ONE

Journal Requirements:

Reviewers' comments:

Reviewer's Responses to Questions

**Comments to the Author**

1. Is the manuscript technically sound, and do the data support the conclusions?

Reviewer #1: Partly

Reviewer #2: Yes

2. Has the statistical analysis been performed appropriately and rigorously? 

Reviewer #1: Yes

Reviewer #2: Yes

3. Have the authors made all data underlying the findings in their manuscript fully available?

Reviewer #1: Yes

Reviewer #2: Yes

4. Is the manuscript presented in an intelligible fashion and written in standard English?

Reviewer #1: No

Reviewer #2: Yes

5. Review Comments to the Author

Reviewer #1: 1. The manuscript requires significant editing for clarity, correct use of language and grammar. I would not recommend accepting this manuscript as currently written.

2. It is unclear to me whether there are only 4 commercial vegan diets in all of Brazil? is this based on a survey of the entire country? this is surprising. Could the authors comment on how common are vegetarian diets ?

3. Food C has vitamin D3 as an ingredient, Food A does not declare whether it has vitamin D2 or D3. Since vitamin D3 is from animal source such as sheep wool, diets that have vitamin D3 are not truly vegan. Therefore this may be a problem in the premise of the study.

4. The sampling amount of each food is not stated.

5. Was a single batch tested from each food? if so, the results may be limited since there may be much batch-to batch variability.

6. The calculation for nutrients per energy requirements as presented is confusing and I am not sure I completely agree with it? I would present the results on a nutrient per 1000 kcal or on DM basis.

7. Since you decided to compare your results to FEDIAF, it is important to consider that some maximums in FEDIAF may be the result of environmental considerations rather than actual data that suggests that this is a physiologic maximum. I believe the zinc maximum is such a value and worth explaining if this is the case (worth checking).

8. The relative activity of vitamin D2 vs. vitamin D3 has been established in cats, and this is worth mentioning.

9. There should be more discussion regarding nutrient digestibility and bioavailability of vegan diets. While not evaluated in the present study, protein and amino acid digestibility and bioavailability may further impact the suitability of vegan diets even if they are analyzed to provide sufficient nutritional content meeting accepted guidelines such as AAFCO or Fediaf.

Reviewer #2: This is an interesting paper which looked at the nutritional adequacy of 4 vegan pet foods currently commercially available in Brazil. The work is relevant and the findings are alarming and need to be published to alert the manufacturers and public alike. In addition given the concern in the pet food industry at the moment around the numbers of dilated cardiac myopathy (DCM) cases associated with grain-free diets formulated with significant amounts of plant proteins this further supports the publication of this manuscript. The work presented here is sound and well written, and the methodology described in the manuscript is accurate and precise. I have attached the manuscript with a few minor suggestions which I hope will improve the manuscript.

General Comment:

The current manuscript just compares the results of the nutrient analyses to FEDIAF guidelines, which are obviously relevant to Europe, but much of the world uses AAFCO nutrient profiles. To increase the reach of the results I would also compare the data with AAFCO profiles. Given the lack of arachidonic acid in any of the diets, they are definitely deficient according to AAFCO also and this should be reported.

Abstract:

I would alter the order of the presentation of the results and talk about the macronutrient first before the micronutrients.

6. PLOS authors have the option to publish the peer review history of their article (what does this mean?). If published, this will include your full peer review and any attached files.

Reviewer #1: No

Reviewer #2: No

---

## [Author Response · Author response to Decision Letter 0]

19 Nov 2019

Dear Editor,

We appreciate your comments and suggestions. They were essential to improve our manuscript. We have made significant changes in the manuscript and attempted to address all comments and concerns.

Please see below the answers in red for each comment. The corrections were inserted in the second version of the manuscript.

Best regards,

Marcio A. Brunetto.

Reviewer #1:

1. The manuscript requires significant editing for clarity, correct use of language and grammar. I would not recommend accepting this manuscript as currently written.

Thank you for the suggestion and based on that, the article was corrected and verified by a scientist proficient in English. All writing and grammar changes have been highlighted in red in the manuscript.

2. It is unclear to me whether there are only 4 commercial vegan diets in all of Brazil? is this based on a survey of the entire country? this is surprising. Could the authors comment on how common are vegetarian diets?

Yes, based on a nationwide survey, only 4 vegan diets are marketed in Brazil. This information has been restated in the text for clarification, and we also added a more detailed description on how the vegan diets were surveyed. Thank you very much.

3. Food C has vitamin D3 as an ingredient, Food A does not declare whether it has vitamin D2 or D3. Since vitamin D3 is from animal source such as sheep wool, diets that have vitamin D3 are not truly vegan. Therefore this may be a problem in the premise of the study.

The ingredient composition present on each vegan food label (according to their manufacturers) has been described. Food A does not state whether it has vitamin D2 or D3 on its label.

In relation to food C, the manufacturer claims to contain vitamin D3. Although vitamin D3 is of animal origin, the product declares itself as vegan commercially. We have emphasized this information and highlight the fact that we analyzed the products that claim to be vegan commercially.

4. The sampling amount of each food is not stated.

We apologize for that, this information has been included.

5. Was a single batch tested from each food? if so, the results may be limited since there may be much batch-to batch variability.

Yes, a batch of each commercial food was analyzed. We understand that there may be a small variation between batches, however this variation may not be large, because the manufacturers maintain a standardization of the ingredients used in food manufacturing as well as the guarantee levels of nutrients. In addition, we understand that the manufacturer must ensure the standard and quality of all manufactured batches of its commercial product, so if one batch has concentrations of some nutrients below the minimum recommendation, it is likely that the other batches will also have. In addition, analyzing several batches of each commercial food would make this study very expensive and therefore unfeasible. Most of the papers published in the literature that analyzed nutrient and / or toxic element concentrations in commercial dog and cat food also analyzed only a single batch of each commercial product. Below are some of these works:

Gray, C. M., Sellon, R. K., & Freeman, L. M. (2004). Nutritional adequacy of two vegan diets for cats. Journal of the American Veterinary Medical Association, 225(11), 1670-1675.

Kanakubo, K., Fascetti, A. J., & Larsen, J. A. (2015). Assessment of protein and amino acid concentrations and labeling adequacy of commercial vegetarian diets formulated for dogs and cats. Journal of the American Veterinary Medical Association, 247(4), 385-392.

da Costa, S. S. L., Pereira, A. C. L., Passos, E. A., Alves, J. D. P. H., Garcia, C. A. B., & Araujo, R. G. O. (2013). Multivariate optimization of an analytical method for the analysis of dog and cat foods by ICP OES. Talanta, 108, 157-164.

Elias, C., De Nadai Fernandes, E., & Bacchi, M. (2011). Neutron activation analysis for assessing chemical composition of dry dog foods. Journal of Radioanalytical and Nuclear Chemistry, 291(1), 245-250.

Fernandes, E. A. D. N., Elias, C., Bacchi, M. A., & Bode, P. (2018). Trace element measurement for assessment of dog food safety. Environmental Science and Pollution Research, 25(3), 2045-2050.

Duran, A., Tuzen, M., & Soylak, M. (2010). Trace element concentrations of some pet foods commercially available in Turkey. Food and Chemical Toxicology, 48(10), 2833-2837.

Kim, H. T., Loftus, J. P., Mann, S., & Wakshlag, J. J. (2018). Evaluation of arsenic, cadmium, lead and mercury contamination in over-the-counter available dry dog foods with different animal ingredients (red meat, poultry, and fish). Frontiers in veterinary science, 5.

Kim, H. T., Loftus, J. P., Gagné, J. W., Rutzke, M. A., Glahn, R. P., & Wakshlag, J. J. (2018). Evaluation of selected ultra-trace minerals in commercially available dry dog foods. Veterinary Medicine: Research and Reports, 9, 43.

Luippold, A., & Gustin, M. S. (2016). Mercury concentrations in wet and dry cat and dog food. Animal Feed Science and Technology, 222, 190-193.

Paulelli, A. C. C., Martins, A. C., de Paula, E. S., Souza, J. M. O., Carneiro, M. F. H., Júnior, F. B., & Batista, B. L. (2018). Risk assessment of 22 chemical elements in dry and canned pet foods. Journal of Consumer Protection and Food Safety, 13(4), 359-365.

Kritikos, G., Weidner, N., Atkinson, J. L., Bayle, J., Van Hoek, I., & Verbrugghe, A. (2018). Quantification of vitamin D3 in commercial dog foods and comparison with Association of American Feed Control Officials recommendations and manufacturer-reported concentrations. Journal of the American Veterinary Medical Association, 252(12), 1521-1526.

Rueckert, C., Braun, C., & Vervuert, I. (2017). Evaluation of nutritional characteristics of commercial canned cat diets. Tierarztliche Praxis. Ausgabe K, Kleintiere/Heimtiere, 45(4), 219-225.

6. The calculation for nutrients per energy requirements as presented is confusing and I am not sure I completely agree with it? I would present the results on a nutrient per 1000 kcal or on DM basis.

Thank you for your comment, we have clarified this matter in the manuscript. The results are also presented in units per 100 grams of dry matter (Tables 3 and 4). However, we also wanted to assess whether the analyzed feeds met the minimum nutrient requirements for animals with low energy requirements. The FEDIAF recommendations in units per 100 grams of dry matter and units per 1000 kcal take into account the amount of food consumed by a dog that has an energy requirement of 95 kcal / kg0.75 and 110 kcal / kg0.75 , and an energy requirement of 75kcal / kg0.67 and 100kcal / kg0.67 for cats. However, FEDIAF itself reports that dogs and cats may have a lower energy requirement of 80kcal / kg0.75 and 52kcal / kg0.67, respectively. There are even studies, which we refer to in the manuscript, that have shown that dogs and cats may have a lower energy requirement. In these cases, the animals will consume a smaller amount of food, so if the concentration of a given nutrient is very close to the minimum recommended, it is possible that the animal would consume an insufficient amount of this nutrient. This was the rationale behind our choice to report the amount of nutrients consumed by a pet with low energy requirements. We understand that this is important, since nowadays it is a reality that animals are more sedentary and(or) castrated, which are factors that result in low energy requirement.

7. Since you decided to compare your results to FEDIAF, it is important to consider that some maximums in FEDIAF may be the result of environmental considerations rather than actual data that suggests that this is a physiologic maximum. I believe the zinc maximum is such a value and worth explaining if this is the case (worth checking).

Thank you for your comment. We had already informed in the manuscript that for copper and zinc the limit is legal, not nutritional. At the end of the manuscript we report that there is not enough information in the literature about excess zinc and copper for dogs and cats, so there is no safe upper limit or nutritional upper limit for these minerals. However, we know that excess transition metals may cause oxidative damage, and this was included in the discussion.

8. The relative activity of vitamin D2 vs. vitamin D3 has been established in cats, and this is worth mentioning.

Thanks for the suggestion. We have included this information in the manuscript.

9. There should be more discussion regarding nutrient digestibility and bioavailability of vegan diets. While not evaluated in the present study, protein and amino acid digestibility and bioavailability may further impact the suitability of vegan diets even if they are analyzed to provide sufficient nutritional content meeting accepted guidelines such as AAFCO or Fediaf.

This was a great suggestion. Thank you. We included a whole paragraph on nutrient digestibility and bioavailability in the discussion.

Reviewer #2:

1. Why not AAFCO 2019?

Thank you, we have included recommendations and comparisons also for AAFCO (2019).

2. Normally ingredients listed by propn of in going weight – this true here?

No, Brazilian law does not require listing of products by order of inclusion, so we do not know the actual order of inclusion of the evaluated commercial products. This information has been added in the manuscript.

We listed ingredients in the order in which the ingredients are in the packaging.

3. Corrected for diets?

The crude fat content of pet foods can be determined with different chemical and physical analysis methods. Soxhlet is the most used methodology for determination of lipids content of a sample, but it depends on the substance that is analyzed. This method consists in the estimation of fat in a sample using the Soxhlet extractor and non-polar solvents like petroleum ether. An alternative for Soxhlet method is using filtering bags technology, developed by Ankom Technology Inc. (Macedon, NY) (LIU, 2011). This alternative methodology was aproved by AOAC in 2005 (AOAC, 2005), and also uses ether, but in a closed system, which decreases extraction time. The cooking process of dry pet food denatures proteins and causes starch gelatinization, which may result in the formation of amylose-lipid complexes which makes it difficult to extract all lipids with ether (LUMLEY; COLWELL, 1991; NRC, 2006). Therefore, the determination of crude fat with acid hydrolysis (CFAH) is recommended for dry pet foods (NRC, 2006). The acid hydrolysis method breaks the amylose-lipid complex and makes it available for fat quantification. Thus, the ethereal extract was performed by the Soxhlet method after acid hydrolysis, the best methodology for pet diets. Please see some references below: 

ASSOCIATION OF THE OFFICIAL ANALYTICAL CHEMISTS (AOAC). Ofﬁcial Methods of Analysis of AOAC International. AOAC International, Gaithersburg, MD, 2005.

LIU, K. Selected topics in the analysis of lipids: modification of an AOCS official method for crude oil content in distillers grains. Urbana: The AOCS Lipid Library, 2011.

LUMLEY, I. D.; COLWELL, R. K. Extraction of fats from fatty foods and determination of fat content. In: ROSSEL, J. B.; PRITCHARD, J. L. R. Analysis of Oilseeds, Fats and Fatty Foods. New York: Elsevier Applied Science. 1991. p. 238–247.

NATIONAL RESEARCH COUNCIL (NRC). Nutrient Requirements of Dogs and Cats. Washington DC, USA, 2006. 398 p. 

4. What?

Sorry, words were missing. The complete phrase is "accelerated cholecystokinin-mediated turnover of bile acids". We made the correction.

---

## [Editor Report · Decision Letter 1]

12 Dec 2019

Nutritional inadequacies in commercial vegan foods for dogs and cats

PONE-D-19-28038R1

Dear Dr. Brunetto,

We are pleased to inform you that your manuscript has been judged scientifically suitable for publication and will be formally accepted for publication once it complies with all outstanding technical requirements.

With kind regards,

Nicoletta Righini, PhD

Academic Editor

PLOS ONE
---

## [Editor Report · Acceptance letter]

7 Jan 2020

PONE-D-19-28038R1 

Nutritional inadequacies in commercial vegan foods for dogs and cats 

Dear Dr. Brunetto:

I am pleased to inform you that your manuscript has been deemed suitable for publication in PLOS ONE. Congratulations! Your manuscript is now with our production department. 

With kind regards,

on behalf of

Dr. Nicoletta Righini 

Academic Editor

PLOS ONE